# Simultaneous Determination and Dietary Risk Assessment of 26 Pesticide Residues in Wheat Grain and Bran Using QuEChERS-UHPLC-MS/MS

**DOI:** 10.3390/foods14244351

**Published:** 2025-12-17

**Authors:** Hongwei Zhang, Quan Liu, Xinhui Dong, Xueyang Qiao, Chunyong Li, Junli Cao, Pengcheng Ren, Jindong Li, Shu Qin

**Affiliations:** 1Shanxi Center for Testing of Functional Agro-Products, Shanxi Agricultural University, Taiyuan 030031, China; 20233460@stu.sxau.edu.cn (H.Z.); 20232342@stu.sxau.edu.cn (Q.L.); 202430460@stu.sxau.edu.cn (X.D.); 202430486@stu.sxau.edu.cn (X.Q.); lichunyong@sxau.edu.cn (C.L.); qinshu@sxau.edu.cn (S.Q.); 2College of Plant Protection, Shanxi Agricultural University, Taiyuan 030031, China

**Keywords:** wheat grain, wheat bran, pesticide residues, dietary risk assessment

## Abstract

Evaluating the potential chronic health risks posed by pesticides to consumers is essential for ensuring food safety and protecting public health. An ultra-high-performance liquid chromatography–tandem mass spectrometry (UHPLC-MS/MS) method coupled with modified QuEChERS extraction was developed to simultaneously determine 26 pesticide residues in wheat grain and bran. Samples were extracted with acetonitrile with 2% (*v*/*v*) acetic acid and cleaned up using C_18_ sorbent. Method validation demonstrated excellent linearity, accuracy, and precision. When applied to 48 wheat grain and 24 bran samples collected from major wheat-growing regions in China, 12 and 21 pesticides were detected at concentrations ranging from <0.005 to 1.785 mg kg^−1^ and <0.01 to 2.188 mg kg^−1^, respectively. Chronic hazard quotients (HQc) and acute hazard quotients (HQa) for all pesticides for grain and bran were far below the safety threshold of 100%. These results indicate that pesticide residues in wheat grain and bran present negligible chronic dietary risks to consumers across all age groups.

## 1. Introduction

Wheat (*Triticum aestivum* L.) remains a strategic crop underpinning Chinese agriculture and global food security. The 2023 agricultural census revealed that China’s wheat cultivation covered 23.62 million hectares, producing 136.6 million metric tons, constituting approximately 20% of the nation’s total grain output [1]. Beyond its dominant caloric contribution (60–75% carbohydrates), wheat delivers 8–15% protein and a spectrum of micronutrients, while its bran fraction supplies additional dietary fibre [2]. Furthermore, wheat bran are integral to vinegar fermentation, animal feed, and functional food applications [3,4]. This economic importance has intensified pesticide-dependent crop protection. The latest national registry lists 4214 authorized formulations for wheat, of which 1488 are composite products [5]. Although these inputs have secured yield gains, persistent residues in wheat grain and bran now constitute a critical quality and safety bottleneck [6]. Chronic exposure has been epidemiologically linked to endocrine and neurotoxic sequelae, with children, pregnant women, and other susceptible cohorts showing heightened vulnerability [7]. Consequently, systematic surveillance of pesticide residues in both wheat grain and bran is mandated to safeguard consumer health.

Although a range of analytical methods has been reported for wheat grain, few studies have simultaneously addressed wheat grain and bran or incorporated probabilistic dietary risk assessment. Wang et al. [8] reported an ultra-performance liquid chromatography–tandem mass spectrometry (UPLC-MS/MS) method for determining imidacloprid and its metabolites in wheat grain, with a limit of quantification (LOQ) of 0.005 mg kg^−1^. Feng et al. [9] used high-performance liquid chromatography–tandem mass spectrometry (LC-MS/MS) combined with the QuEChERS method to quantify trifloxystrobin, trifloxystrobin acid, and tebuconazole in wheat grain, observing terminal residues <0.001 mg kg^−1^. Kolberg et al. [10] systematically evaluated 24 pesticides in wheat grain using the QuEChERS method coupled with gas chromatography–mass spectrometry (GC-MS), achieving excellent linearity, reliable accuracy, and precision. For the risk assessment of pesticides in wheat grain, Qin et al. [11] calculated chronic hazard quotients of 0.0014% and 0.031% for mesosulfuron-methyl and diflufenican, respectively, whereas Yang et al. [12] evaluated the risk of cycloxydim in different age groups, ranging from 0.33 to 0.81%. Yigitsoy et al. [13] investigated pesticide residues in the major wheat-producing regions of Turkey. Their results indicated that both the chronic and acute health risk quotients (HQc and HQa) for adults and children were below the threshold of 1. In a study of 80 wheat samples from Algeria, Mebdoua et al. [14] found that 5% of samples exceeded the maximum residue limits (MRLs) for benalaxyl, chlorpyrifos, and metalaxyl; however, the estimated long-term exposure for these pesticides remained below the acceptable daily intake (ADI). Kalantary et al. [15] analyzed pesticide levels in wheat grains and their processed products collected from 12 fields in the Arak-Bidgol region of Iran. The results demonstrated that the total health risks for both children and adults were less than 1. Yet, these assessments were confined to single actives and do not consider wheat bran, the fraction most exposed to field sprays and post-harvest treatments.

Recent evidence indicates pronounced pesticide enrichment in wheat bran. Kolberg et al. [10] observed that 24 pesticides residue were three times higher in wheat bran. Teló et al. [16] reported that thiamethoxam residues were 5.8 times higher in rice bran than in the endosperm. Despite this evidence, the global maximum residue limit (MRL) was set only for wheat grain; regulatory guidance and quantitative dietary risk data for wheat bran were largely absent.

To address these gaps, an ultra-high-performance liquid chromatography–tandem mass spectrometry (UHPLC-MS/MS) method, based on an optimized QuEChERS extraction procedure, was developed to detect 26 pesticides in both wheat grain and bran. The method was applied to 48 wheat grain and 24 bran samples collected from major Chinese wheat-growing regions. Residue datasets were integrated into probabilistic dietary exposure models stratified by six age groups. The resulting risk estimates provide the quantitative foundation required to refine the maximum residue limit for bran and to protect vulnerable consumer populations.

## 2. Materials and Methods

### 2.1. Materials and Reagents

Certified reference standards for the 26 target pesticides (1000 mg L^−1^ in acetonitrile, purity > 98.2%) were obtained from the Quality Supervision, Inspection, and Testing Center of the Agro-Environmental Protection Institute, Ministry of Agriculture and Rural Affairs (Tianjin, China). HPLC-grade acetonitrile and formic acid were purchased from Fisher Scientific (Waltham, MA, USA); HPLC-grade acetic acid was supplied by Tianjin Damao Chemical Reagent Factory (Tianjin, China). Analytical-grade anhydrous magnesium sulfate (MgSO_4_) and sodium chloride (NaCl) were obtained from Sinopharm Chemical Reagent Co., Ltd. (Shanghai, China). Primary secondary amine (PSA, 40–60 μm) and octadecylsilyl (C_18_, 40–60 μm) were acquired from Bonna-Agela Technologies (Tianjin, China). Ultrapure water was produced by Guangzhou Watson’s Food & Beverage Co., Ltd. (Guangzhou, China), and 0.22 μm syringe filters were sourced from Peak Sharp Technologies (Beijing, China).

### 2.2. Standard Solution Preparation

Stock solutions of the 26 pesticides (1000 mg L^−1^) were individually prepared in 10 mL volumetric flasks and diluted to volume with acetonitrile. Matrix-matched calibration standards were prepared using blank wheat grain and bran extracts to account for matrix effects during analysis. All solutions were stored at 4 °C until use.

### 2.3. Instrumentation and Chromatographic Conditions

Quantitative determination was carried out on a Waters Xevo TQ-XS triple-quadrupole tandem mass spectrometer coupled to an ACQUITY UPLC I-Class system (Waters, Milford, MA, USA). Chromatographic separation was achieved on an ACQUITY UPLC BEH C_18_ column (100 mm × 2.1 mm, 1.7 µm) thermostated at 40 °C. The mobile phase consisted of (A) 0.2% (*v*/*v*) formic acid in water and (B) acetonitrile, delivered at 0.30 mL min^−1^ with an injection volume of 2 µL. A 6 min gradient elution programme was applied: 0–1.5 min, 90 → 50% A; 1.5–3.5 min, 50% A; 3.5–4.0 min, 50 → 10% A; 4.0–5.0 min, 10% A; 5.0–5.5 min, 10 → 90% A; 5.5–6.0 min, 90% A.

Mass spectrometric detection was performed in positive electrospray ionization (ESI^+^) mode using multiple reaction monitoring (MRM). The optimized source parameters were as follows: capillary voltage, 2.5 kV; source temperature, 400 °C; desolvation gas flow, 800 L h^−1^. Compound-specific MRM transitions and collision energies were summarized in Table 1.

### 2.4. Sample Pre-Treament

For sample preparation, 5.00 g of wheat grain or 2.50 g of bran were weighed into separate 50 mL centrifuge tubes. Subsequently, 5 mL of water and 10 mL of acetonitrile with 2% (*v*/*v*) acetic acid were added to each tube. The mixtures were vortexed and shaken for 10 min. Next, 3 g of anhydrous MgSO_4_ and 2 g of NaCl were added, followed by vigorous shaking at 2500 rpm for 10 min. The samples were then centrifuged at 8000 rpm for 5 min.

After centrifugation, 1.5 mL of the supernatant was transferred to a 2 mL centrifuge tube containing 150 mg of anhydrous MgSO_4_ and 100 mg of C_18_. This mixture was vortexed and centrifuged again at 8000 rpm for 3 min. Finally, the extracts were filtered through a 0.22 μm syringe filter prior to analysis by UHPLC-MS/MS.

### 2.5. Method Validation

The validation process followed SANTE/11312/2021 guidelines [17], including linearity, accuracy, precision, sensitivity, and matrix effect (ME).

#### 2.5.1. Linearity and Limit of Quantification

Solvent-based calibration standards were prepared by serially diluting the 26 pesticides stock solution in acetonitrile to achieve concentrations ranging from 0.0025 to 0.2 mg L^−1^ for both wheat grain and bran. Matrix-matched standards at different levels were prepared using blank wheat grain and grain bran extracts processed as described in Section 2.4. Calibration curves for wheat grain and wheat bran were constructed using nine-point serial dilutions. The concentrations for the standards were as follows: 0.001, 0.002, 0.0025, 0.005, 0.01, 0.02, 0.05, 0.1, and 0.2 mg kg^−1^ for wheat grain and 0.002, 0.0025, 0.005, 0.01, 0.02, 0.025, 0.05, 0.1, and 0.2 mg kg^−1^ for wheat bran. Linearity was assessed by constructing calibration curves with pesticide concentration (*x*-axis) plotted against chromatographic peak area (*y*-axis). The limit of quantification (LOQ) was determined as the lowest validated spiking level.

#### 2.5.2. Precision and Accuracy

Blank wheat grain and bran samples were spiked with standard solutions of 26 pesticides at three concentrations: 0.005, 0.1, and 0.2 mg kg^−1^ for wheat grain; 0.01, 0.1, and 0.2 mg kg^−1^ for bran, with five replicates per concentration. Recovery and relative standard deviation (RSD) were calculated for each analyte.

#### 2.5.3. Matrix Effect

ME arises from non-analyte components in the sample matrix, which may interfere with analytical accuracy [18], and was calculated as follows:(1)$$\mathrm{ME}=\left(\frac{A}{B}-1\right)\times 100\%$$
where A is the slope of the matrix-matched standard solution, and B is the slope of the acetonitrile. −20% < ME < 20%: Weak (negligible) matrix effect; −50% ≤ ME ≤ −20% or 20% ≤ ME ≤ 50%: Moderate matrix effect; ME < −50% or ME > 50%: Strong matrix effect [19].

### 2.6. Sample Collection

Wheat grain samples were systematically collected from 12 regions across China and wheat bran samples from 11 regions. The wheat grain samples included Shanxi Province, Beijing Municipality, Heilongjiang Province, Hunan Province, Anhui Province, Henan Province, Ningxia Hui Autonomous Region, Qingdao (Shandong Province), Taian (Shandong Province), Shanghai Municipality, Yunnan Province, and Chongqing Municipality. The wheat bran samples were from Hengshui (Hebei Province), Shijiazhuang (Hebei Province), Zhoukou (Henan Province), Zhumadian (Henan Province), Anyang (Henan Province), Jining (Shandong Province), Linyi (Shandong Province), Huainan (Anhui Province), Huaiyang (Anhui Province), Lianyungang (Jiangsu Province), and Taiyuan (Shanxi Province). With the exception of four bran samples from Shanxi Province, all wheat grain and bran samples were collected in duplicate (two per region) during 2024. The wheat grains were procured directly from farmers’ fields, whereas the bran was obtained from local mills. Upon arrival at the laboratory, the wheat grains were ground into a fine powder using a Chinese herbal medicine grinder. All samples were cryopreserved at −20 °C in vacuum-sealed polyethylene bags to ensure chemical stability until analysis.

### 2.7. Health Risk Assessment

#### 2.7.1. Chronic Risk Assessment

Chronic hazard quotients (HQc) was evaluated by comparing the national estimated daily intake (NEDI) with the acceptable daily intake (ADI), calculated as follows:(2)$$\mathrm{NEDI}=\frac{\sum \left(\mathrm{MR}i\;\times \;\mathrm{Fi}\right)}{\mathrm{bw}}$$(3)$$\mathrm{HQc}=\frac{\mathrm{NEDI}}{\mathrm{ADI}}\;\times \;100\%$$
where NEDI is the national estimated daily intake (mg kg^−1^ bw) sourced from the Ministry of Agriculture of the People’s Republic of China Notice No. 2308 [20]; MRi is the mean residue level, mg kg^−1^; Fi is the dietary amount of wheat grain, and the data were obtained from Report on Nutrition and Health Status of Chinese Residents [21], g d^−1^; ADI is the acceptable daily intake, mg kg^−1^ bw; bw is the body weight, kg. HQc < 100% and HQc > 100% reflect acceptable and unacceptable chronic risk, respectively [22].

#### 2.7.2. Acute Risk Assessment

Acute hazard quotients (HQa) were based on the International Estimated Short-Term Intake (IESTI) and the Acute Reference Dose (ARfD), calculated as follows:(4)$$\mathrm{IESTI}=\frac{LP\times HR}{bw}$$(5)$$\mathrm{HQa}=\frac{\mathrm{IESTI}}{\mathrm{ARfD}}\times 100\%$$
where IESTI is the international estimated short-term intake (mg kg^−1^ bw) derived from the World Health Organization [23]; The large portion (LP, 97.5th percentile) consumption data were obtained from the WHO food consumption database [24], g d^−1^. The highest residue level (HR) is the highest residue level, mg kg^−1^; The acute reference dose (ARfD) was sourced from the JMPR report [25], mg kg^−1^ bw. HQa < 100% and HQa > 100% reflect acceptable and unacceptable acute risk, respectively [26].

## 3. Results and Discussion

### 3.1. Optimization of Extraction Procedure

QuEChERS (Quick, Easy, Cheap, Effective, Rugged, and Safe) sample pretreatment [27] has emerged as the gold standard for pesticide residue analysis in complex food matrices in recent years [28]. Given compositional differences between wheat grain and bran, we modified the QuEChERS pretreatment separately for each matrix. Based on previous reports [29,30] three acetonitrile-based extraction solvents were systematically evaluated for 26 target pesticides: (A1) acetonitrile, (A2) acetonitrile with 1% (*v*/*v*) acetic acid, and (A3) acetonitrile with 2% (*v*/*v*) acetic acid. Appropriate acetonitrile acidification treatment facilitates the dissolution of the target compound from the sample matrix and enhances extraction efficiency [30]. For wheat grain, A1 and A2 provided recovery of 70.6–105.1% and 86.3–125.7%, but RSD was unacceptable. The RSD of metsulfuron-methyl and flumetsulam were 25.5–41.9%. A3 gave the best recovery (91.4–108.3%) with RSD less than 12.3% (Figure 1A), likely due to their interaction with wheat grain proteins [28,31]. Similar trends were observed in bran samples, where A3 consistently provided recovery of 73.2–105.1% with excellent precision (RSD: 0.9–7.0%) (Figure 2A), following the requirements of SANTE/11312/2021 guidelines. Based on these results, A3 was selected for all subsequent extractions of both wheat grain and bran.

Salt, typically composed of NaCl and MgSO_4_, enhances pesticide transfer to the organic phase by decreasing their aqueous solubility, thereby improving extraction efficiency [26]. In this study, two salt formulations were systematically evaluated to identify the optimal composition for the target pesticides. Formulation B1 (4 g MgSO_4_ + 1 g NaCl) and formulation B2 (3 g MgSO_4_ + 2 g NaCl) were compared. For wheat grain, B2 exhibited high extraction efficiency, with recovery ranging from 92.3 to 109.0% with RSD of 0.7–1.5% (Figure 1B). In bran, both formulations delivered comparable recovery; however, B2 provided slightly superior precision, with RSD of 0.6–13.6% (Figure 2B). Thus, B2 was selected as salt for both wheat grain and bran.

The extraction process of wheat grain and bran inevitably co-extracts matrix components, such as lipids, pigments, and sugars [32], which may interfere with subsequent analyses and contaminate the LC-MS/MS system. PSA was used to adsorb organic acids, sugars, phenols, and pigments, while C_18_ could strongly adsorb non-polar compounds and lipids [33]. In this study, 150 mg anhydrous MgSO_4_ was used to remove the residual water and the purification effect of three different clean-up adsorbents was investigated: (C1) 50 mg PSA + 50 mg C_18_, (C2) 100 mg C_18_, and (C3) 100 mg PSA. For wheat grain, C1 and C3 showed unsatisfactory recovery (50.4–60.2%) for metsulfuron-methyl and flumetsulam, consistent with previous reports which found that PSA induced analyte loss [34]. Conversely, C2 demonstrated superior performance for wheat grain, with recovery ranging from 91.4 to 109.0% and RSD ≤ 11.5% (Figure 1C). Similar patterns emerged in bran, where C2 again provided the most consistent results, where the recovery was 80.6–110.0% and RSD was 1.1–11.2% (Figure 2C). Consequently, C2 was selected as the primary clean-up sorbent for subsequent analyses.

### 3.2. Method Validation

Method validation was performed in accordance with SANTE/11312/2021 guidelines [17] and encompassed linearity, limit of quantification, accuracy, precision, and ME.

As shown in Table 2, all 26 pesticides exhibited excellent linearity in both wheat grain and bran, with correlation coefficients (R^2^) ≥ 0.9949. The LOQ, defined as the lowest spiking level, was determined to be 0.005 mg kg^−1^ for wheat grain and 0.01 mg kg^−1^ for bran.

ME, which can enhance or suppress the signal of the target analyte due to co-extracted substances, are common in UHPLC-MS/MS analysis and can impact analytical accuracy [35]. In wheat grain, ME ranged from −91.83% to 70.51%, whereas in bran they ranged from −72.92% to 62.85% (Table 2). There were 21 pesticides (80.8%) that exhibited absolute ME > 20%, indicating moderate to strong matrix interference. Consequently, matrix-matched calibration curves were utilized in this study to enable accurate quantitative analysis and minimize ME.

In wheat grain, the recovery of 26 pesticides at three concentration levels ranged from 73.2% to 116.8%, with RSD below 17.9%. For wheat bran, the recovery were 87.7–110.7%, with RSD ranging from 0.8% to 18.5%. The detailed results are presented in Table 3. These data demonstrate compliance with established acceptance criteria for pesticide residue analytical methods [17].

### 3.3. Application to Real Samples

The established method was applied to analyze 48 wheat grain and 24 bran samples collected from major producing regions of China. The results are presented in Figure 3 and Appendix A.

In wheat grain, 12 pesticides were detected, comprising 3 insecticides, 7 fungicides and 2 herbicides (Figure 3A). Azoxystrobin, a broad-spectrum strobilurin fungicide, was the most prevalent compound, being detected in 93.8% of the samples at concentrations ranging from 0.005 to 0.3925 mg kg^−1^; all values remained below the Chinese MRL of 0.5 mg kg^−1^. These levels align with recent findings in wheat–maize rotations, where azoxystrobin grain residues remained <0.004 mg kg^−1^ [36]. Triazole fungicides epoxiconazole followed, detected in 31.3% of samples, whereas the strobilurin kresoxim-methyl was found in 18.8%. Regional exceedances of national MRL were observed. Epoxiconazole concentrations in samples originating from Beijing and Shanxi exceeded the Chinese MRL of 0.05 mg kg^−1^, an outcome that previous chamber and field studies attribute to enhanced acropetal translocation under elevated temperature and humidity regimes [37,38]. Similarly, kresoxim-methyl residues in samples from Heilongjiang and Shanxi surpassed the MRL of 0.05 mg kg^−1^. Comparable regional heterogeneity has been reported for rice matrices, where climate-driven differences in application timing, soil sorption, and post-harvest handling explained 36% of MRL violations [39,40]. For difenoconazole, the results were similar to those for fenpyroximate, with concentrations below 0.01 mg kg^−1^ detected at the recommended dose (135 g a.i. hm^−2^), making it suitable for use in field applications [41]. Additionally, multiple triazole fungicides (epoxiconazole, tebuconazole, and tricyclazole) have been detected in commercial samples [42]. In contrast, more lipophilic difenoconazole and strobilurins fungicides (azoxystrobin, kresoxim-methyl and trifloxystrobin) were predominantly concentrated in lipid-rich fractions, particularly paddy, bran, and brown rice, resulting in detectable residues in these matrices [42]. Among the remaining analytes, the neonicotinoid insecticide clothianidin and the sulfonylurea herbicide bensulfuron methyl were each detected in 16.7% of the samples, followed by tebuconazole (10.4%), the neonicotinoid imidacloprid (6.3%), and the phenylurea herbicide chlortoluron (2.1%). Critically, tebuconazole, bensulfuron methyl, and metsulfuron-methyl exceeded their respective MRL in specific provinces: tebuconazole in Beijing and Yunnan and both sulfonylureas in Chongqing. Although these herbicides were applied at ultra-low doses [43], their limited volatility and strong soil sorption promote accumulation under the intensive wheat–rice rotations typical of southern China, increasing the risk of phytotoxic carry-over to subsequent crops [44,45].

In the bran samples, 21 pesticides were detected, comprising 7 insecticides, 8 fungicides, and 6 herbicides (Figure 3B). Both the number of detected pesticides and their residue levels exceeded those found in wheat grain. The neonicotinoid clothianidin displayed the highest incidence (83.3%), followed by the triazole tebuconazole and strobilurin azoxystrobin, which were each detected in 66.7% of the samples, and the neonicotinoid imidacloprid (62.5%). Tebuconazole residues spanned <0.010–2.19 mg kg^−1^, primarily in samples from Hengshui (Hebei), Zhoukou (Henan), Jining (Shandong), and Huaiyang (Anhui). These elevated levels were consistent with residue dynamics studies showing that reduced precipitation and elevated post-application temperatures prolong foliar half-life and enhance penetration into the outer grain layers [46,47]. The high detection rate of clothianidin may be due to its accumulation in bran, which was highly lipophilic [48]. Yan et al. [49] reported that epoxiconazole, applied at recommended and double field rates, persisted at 0.14–2.54 mg kg^−1^ in rice husks even 28 days after the last treatment. The present data confirmed similar retention in wheat bran, underscoring the role of the aleurone layer as a sink for moderately lipophilic triazoles. Other frequently detected compounds included epoxiconazole and hexaconazole (50.0%), difenoconazole and thiamethoxam (45.8%), dimethoate (37.5%), and flumetsulam (25.0%). Lowest frequencies were recorded for isoproturon, acetamiprid, dichlorvos (20.8%), and diflufenican (8.3%). Pesticide residues in bran were consistently higher than those in wheat grain, likely due to bran’s prolonged exposure and direct contact with pesticides during processing [50,51].

### 3.4. Dietary Exposure Assessment

#### 3.4.1. Chronic Risk Assessment

Chronic dietary risk of 12 pesticides quantified in wheat grain (n = 48) and 21 pesticides in wheat bran (n = 24) was assessed. Chronic dietary risk refers to health hazards from prolonged pesticide residue exposure over a lifetime [35]. Chronic dietary risk assessment combines maximum residue limit with the national estimated daily intake and chronic hazard quotients. The Chinese population was stratified into 11 subgroups by age and sex: males and females aged 0–35 months, 3–5 years, 6–14 years, 15–49 years, 50–74 years, and ≥75 years. HQc were calculated at the 50th, 90th, 95th, and 97.5th percentiles of exposure.

For wheat grain, HQc values across all 12 pesticides were less than 2.48% (Figure 4), indicating negligible lifetime risk. The highest HQc values were observed in the 0–35 months cohort, likely due to lower body weights, with a progressive decrease across older age groups [52]. Males exhibited higher HQc values than females in the 3–5 years, 50–74 years, and ≥75 years groups, whereas females showed elevated values in the 6–14 years and 15–49 years cohorts. Hexaconazole (Figure 4L) displayed the highest HQc values across all age groups, attributable to its low ADI (0.005 mg kg^−1^ bw), which increases relative risk. HQc values for all detected pesticides ranged from 0.0001% to 2.4799%, all below the 100% threshold. Although kresoxim-methyl, metsulfuron-methyl, bensulfuron methyl, tebuconazole, and epoxiconazole residues sporadically exceeded the corresponding Chinese MRL, their HQc values were less than 0.6%, confirming an acceptable chronic risk.

Dietary intake was intrinsically linked to health outcomes. Bran, commonly used as a raw material in food processing and daily diets, poses a potential dietary exposure risk, making its safety assessment crucial. National food consumption data indicated that bran intake is negligible before 6 years; therefore, the deterministic risk assessment was restricted to children (6–14 years), adults (15–49 years), and older adults (50–74 years), disaggregated by sex. The HQc values of wheat bran are shown in Figure 5. Across all percentiles, HQc values increased monotonically with exposure level and were consistently highest among 6–14 year-old males, consistent with their elevated bran consumption rates [53]. Tebuconazole, with the lowest ADI (0.03 mg kg^−1^ bw), drove the greatest lifetime risk (HQc range from 0.0954 to 6.1171%). It was followed by dimethoate (0.0703–4.5045%) and hexaconazole (0.0287–1.378%). Despite these maxima, all HQc values remained less than 100%, indicating that the chronic intake of bran containing the investigated pesticide residues poses no appreciable health threat for any demographic subgroup.

#### 3.4.2. Acute Risk Assessment

Acute dietary risk was performed for pesticides detected in wheat grains and wheat bran, with 5 and 10 pesticides identified in the two matrices, respectively. As illustrated in Figure 6A, the acute Hazard Quotient (HQa) values for all pesticides in wheat grains were below 2.4171%. Notably, while the residue level of tebuconazole exceeded the maximum residue limit (MRL) established in China, its calculated HQa values ranged from 0.2364 to 2.4171%. As all values were substantially below 100%, the acute dietary exposure to these pesticides, including tebuconazole, does not pose a significant short-term health risk to consumers. The acute hazard quotient (HQa) values for all pesticides detected in wheat bran were below 12.3603% (Figure 6B). Among these, diazinon presented the highest acute dietary risk, with HQa values ranging from 3.8111 to 12.3603%. This may be attributed to the high detection rate (0.4278 mg kg^−1^) coupled with the relatively low ARfD of 0.03 mg kg^−1^ bw. Dimethoate (HQa: 3.4125–11.0676%), tebuconazole (HQa: 2.0262–6.5715%), and dichlorvos (HQa: 1.3906–4.5009%) followed in descending order of risk, while the HQa values of the remaining pesticides ranged from 0.0093 to 0.7351%. Notably, three of the detected pesticides in wheat bran belonged to organophosphorus pesticides (OPPs). These pesticides are highly toxic and bioaccumulative, posing a serious threat to human and wildlife health [54]. In larval zebrafish, moderate exposure to organophosphate pesticides (OPPs) is commonly associated with muscle paralysis and a reduced heart rate, while severe exposure can also result in decreased trunk length [55]. An analysis of OPP residues in the daily food intake of the Hungarian population revealed that over 99.95% of the estimated daily residual intake remained below the acute reference dose [56], a finding consistent with our results. All calculated Hazard Quotient (HQa) values were below 100%, indicating that the short-term ingestion of bran-containing residues of the studied pesticides does not pose a significant health risk.

## 4. Discussion

This study developed a modified QuEChERS-UHPLC-MS/MS to simultaneously determine 26 pesticide residues in wheat grain and bran. This method demonstrated excellent extraction efficiency, with recovery ranging from 73.2% to 116.8% and RSD between 0.8% and 18.5%. When integrated with the validated UHPLC-MS/MS method, the LOQ were as low as 0.005 mg kg^−1^ for wheat grain and 0.01 mg kg^−1^ for bran. Analysis of wheat grain and bran samples revealed the detection of 12 pesticides in the grain and 21 pesticides in the bran. The concentration ranges were 0.005–1.785 mg kg^−1^ and 0.01–2.188 mg kg^−1^, with detection rates of 46.2% and 80.8%, respectively. Residue levels of pesticides detected in bran consistently exceeded those in wheat grain. Chronic and acute dietary risk assessments for different population groups, based on pesticides detected in both wheat and bran, yielded chronic hazard quotients and acute hazard quotients all below 100%, indicating no unacceptable risk to human health. The proposed method is rapid, sensitive, and precise, making it suitable for routine multi-residue screening in cereal matrices. Additionally, it provides reliable data for dietary risk assessments of cereal-based foods.

## Figures and Tables

**Figure 1 foods-14-04351-f001:**
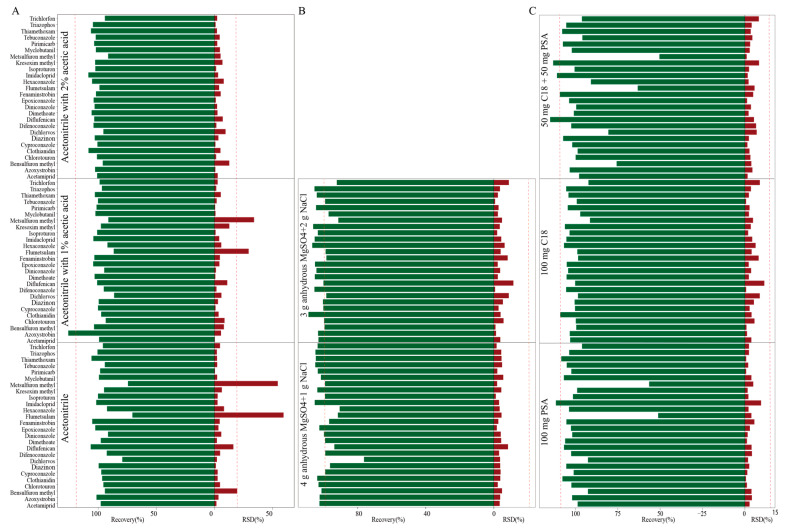
Effect of extraction solvent (A), salt (B), and clean-up adsorbents (C) for 26 pesticides in wheat grain.

**Figure 2 foods-14-04351-f002:**
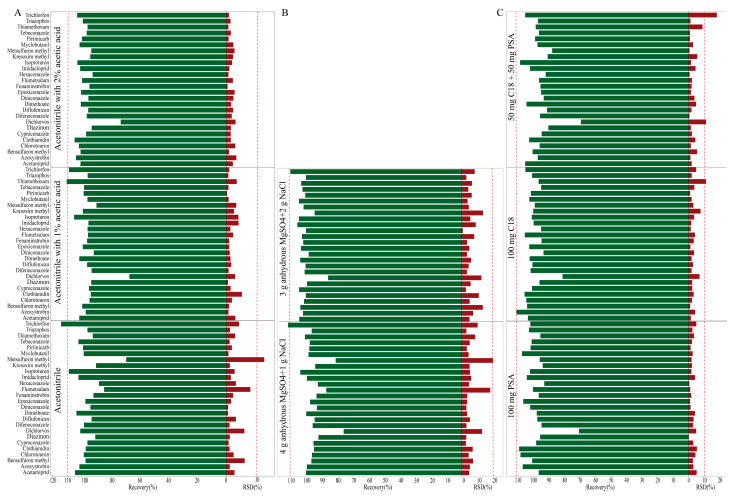
Effect of extraction solvent (**A**), salt (**B**), and clean-up adsorbents (**C**) for 26 pesticides in bran.

**Figure 3 foods-14-04351-f003:**
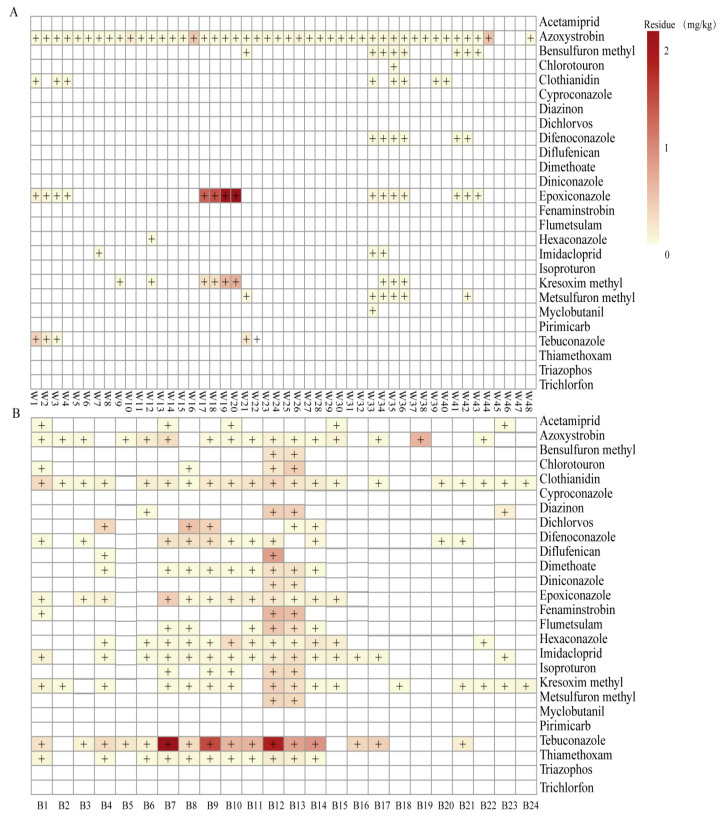
Colour map of pesticides detected in wheat grain (**A**) and bran (**B**) samples. White: <LOQ. The darker the colour, the higher the residue. Note: “+” indicates positive sample.

**Figure 4 foods-14-04351-f004:**
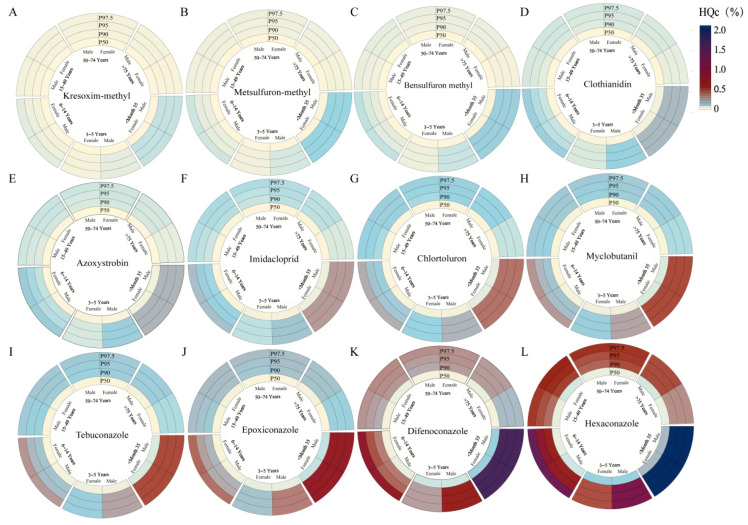
Chronic dietary risk assessment of pesticide in wheat grain at different ages. P50–P97.5 refers to various percentiles of the probability model. Note: Panels (**A**–**L**) represent the following, respectively: (**A**) Kresoxim-methyl; (**B**) Metsulfuron-methyl; (**C**) Bensulfuron methyl; (**D**) Clothianidin; (**E**) Azosystrobin; (**F**) Imidacloprid; (**G**) Chlortoluron; (**H**) Myclobutanil; (**I**) Tebuconazole; (**J**) Epoxiconazole; (K) Difenoconazole; (**L**) Hexaconzole.

**Figure 5 foods-14-04351-f005:**
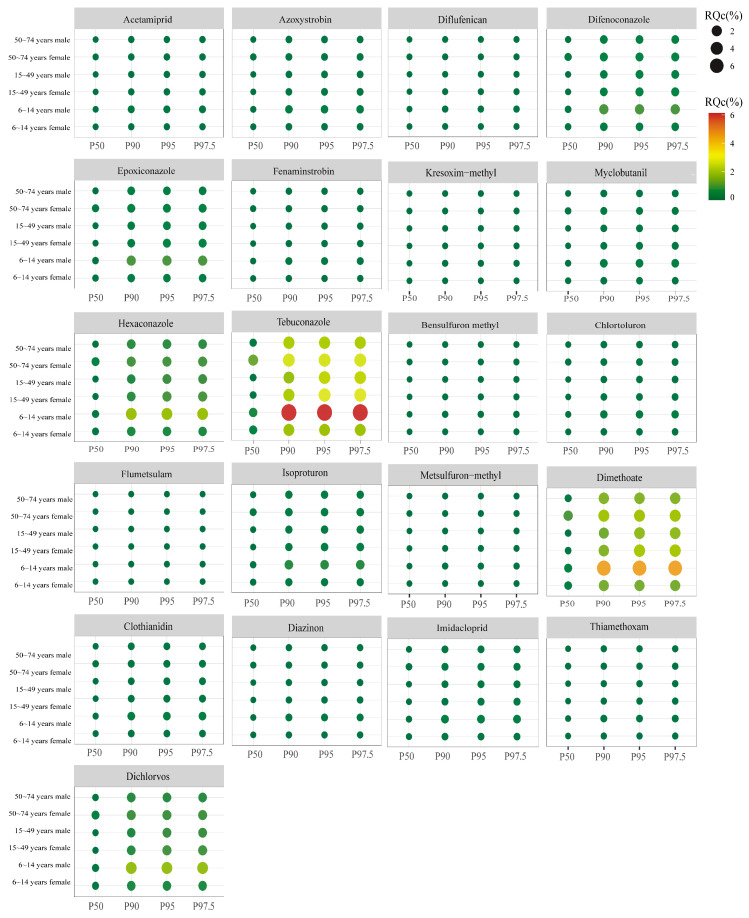
Chronic dietary risk of pesticide bran at different ages. P50–P97.5 refers to varying percentiles of the probability model.

**Figure 6 foods-14-04351-f006:**
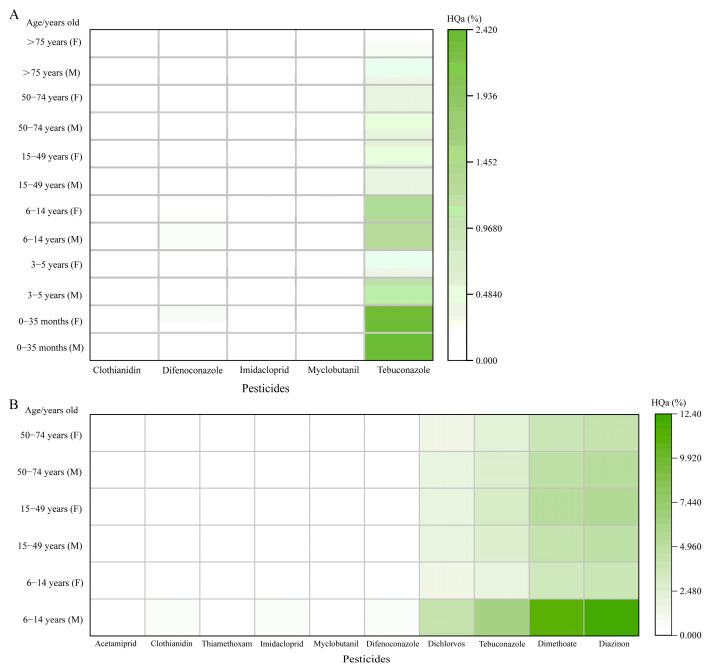
Acute dietary risk of pesticide wheat grain and wheat bran at different ages (**A**,**B**).

**Table 1 foods-14-04351-t001:** Multiple reaction monitoring (MRM) parameters for 26 pesticides.

Pesticides	Type	Molecular Formula	Retention Time (min)	Precursor Ion (*m*/*z*)	Product Ion (*m*/*z*)	Cone Voltage (V)	Collision Energy (eV)
Acetamiprid	Insecticide	C_10_H_11_ClN_4_	2.54	223	56.1/126 *	34	15/20
Azoxystrobin	Fungicide	C_22_H_17_N_3_O_5_	4.4	404	329/372 *	28	30/15
Bensulfuron methyl	Herbicide	C_16_H_18_N_4_O_7_S	3.44	411.1	149 */128	30	22/20
Chlortoluron	Herbicide	C_10_H_13_ClN_2_O	3.12	213	46/72 *	25	15/15
Clothianidin	Insecticide	C_6_H_8_ClN_5_O_2_S	2.47	250	132/169 *	30	18/12
Cyproconazole	Fungicide	C_15_H_18_CIN_3_O	4.11	292.2	70.2 */125.1	36	18/24
Diazinon	Insecticide	C_12_H_21_N_2_O_3_PS	5.13	305.1	96.9/169 *	31	35/22
Dichlorvos	Insecticide	C_4_H_7_Cl_2_O_4_P	2.90	221	109 */79	20	15/25
Difenoconazole	Fungicide	C_19_H_17_Cl_2_N_3_O_3_	5.03	406	111.1/251.1 *	37	60/25
Diflufenican	Herbicide	C_19_H_11_F_5_N_2_O_2_	5.18	359.1	266 */246	26	24/32
Dimethoate	Insecticide	C_5_H_12_NO_3_PS_2_	2.54	230.1	125 */199	40	20/10
Diniconazole	Fungicide	C_15_H_17_Cl_2_N_3_O	4.91	326.1	70.2 */159	46	25/34
Epoxiconazole	Fungicide	C_17_H_13_ClFN_3_O	4.51	330	121.04/101 *	40	20/40
Fenaminstrobin	Fungicide	C_17_H_19_NO_4_	5.19	434	171 */212	10	17/15
Flumetsulam	Herbicide	C_12_H_9_F_2_N_5_O_2_S	2.58	326.1	109/129 *	37	50/25
Hexaconazole	Fungicide	C_14_H_17_Cl_2_N_3_O	4.83	314	70.1*/159	40	22/28
Imidacloprid	Insecticide	C_9_H_10_ClN_5_O_2_	2.51	256.1	175.1 */209.1	30	20/15
Isoproturon	Herbicide	C_12_H_18_N_2_O	3.28	207	47/72 *	30	16/22
Kresoxim-methyl	Fungicide	C_18_H_19_NO_4_	4.99	314.1	116 */206	15	12/7
Metsulfuron-methyl	Herbicide	C_14_H_15_N_5_O_6_S	2.87	411.2	167 */198.9	30	30/15
Myclobutanil	Fungicide	C_15_H_17_ClN_4_	4.50	289.1	70.2 */125.1	34	18/32
Pirimicarb	Insecticide	C_11_H_18_N_4_O_2_	2.41	239.1	72 */182.1	30	18/15
Tebuconazole	Fungicide	C_16_H_22_ClN_3_O	4.71	308	70.1 */125	40	22/40
Thiamethoxam	Insecticide	C_8_H_10_ClN_5_O_3_S	2.40	292	132/211.2 *	40	20/12
Triazophos	Insecticide	C_12_H_16_N_3_O_3_PS	4.81	314.1	118.9 */161.9	31	35/18
Trichlorfon	Insecticide	C_4_H_8_Cl_3_O_4_P	2.44	257	79/109 *	28	30/18

*: Quantitative transition.

**Table 2 foods-14-04351-t002:** Calibration equation, correlation coefficient (R^2^), and matrix effect (ME) of 26 pesticides.

Analyte	Solvent		Wheat Grain			Wheat Bran		
	Calibration Curve	R^2^	Calibration Curve	R^2^	ME (%)	Calibration Curve	R^2^	ME (%)
Acetamiprid	y = 1.925 × 10^6^x + 7.200 × 10^2^	0.9999	y = 1.724 × 10^6^x + 1.654 × 10^3^	0.9996	−10.47	y = 1.310 × 10^6^x + 3.007 × 10^3^	0.9997	−31.95
Azoxystrobin	y = 5.247 × 10^6^x + 8.730 × 10^2^	0.9999	y = 7.109 × 10^6^x + 8.519 × 10^3^	0.9995	35.49	y = 8.545 × 10^6^x + 2.382 × 10^4^	0.9996	62.85
Bensulfuron methyl	y = 2.831 × 10^6^x − 5.950 × 10^2^	0.9999	y = 3.701 × 10^6^x + 2.433 × 10^3^	0.9999	30.71	y = 4.410 × 10^6^x − 1.394 × 10^3^	0.9997	55.78
Chlortoluron	y = 6.238 × 10^6^x − 2.390 × 10^2^	0.9998	y = 6.382 × 10^6^x + 3.756 × 10^3^	0.9996	1.57	y = 5.181 × 10^6^x − 4.120 × 10^3^	0.9995	−17.00
Clothianidin	y = 4.555 × 10^5^x + 1.849 × 10^5^	0.9997	y = 3.228 × 10^5^x + 7.610 × 10^2^	0.9997	−29.14	y = 2.198 × 10^5^x + 1.759 × 10^3^	0.9988	−51.75
Cyproconazole	y = 6.546 × 10^6^x + 2.829 × 10^3^	0.9999	y = 6.128 × 10^6^x + 4.475 × 10^3^	0.9999	−6.38	y = 5.398 × 10^6^x + 1.100 × 10^3^	0.9998	−17.54
Diazinon	y = 4.104 × 10^7^x − 1.343 × 10^5^	0.9998	y = 3.354 × 10^6^x + 2.383 × 10^4^	0.9999	−91.83	y = 2.001 × 10^7^x + 1.135 × 10^4^	0.9996	−51.23
Dichlorvos	y = 9.001 × 10^6^x + 1.826 × 10^5^	0.9960	y = 7.693 × 10^6^x + 1.150 × 10^3^	0.9990	22.44	y = 6.354 × 10^6^x + 3.203 × 10^4^	0.9975	1.13
Difenoconazole	y = 7.124 × 10^6^x − 1.078 × 10^4^	0.9999	y = 5.542 × 10^6^x + 1.898 × 10^3^	0.9997	−22.21	y = 1.929 × 10^6^x + 5.26 × 10^2^	0.9998	−8.26
Diflufenican	y = 1.306 × 10^7^x − 4.901 × 10^3^	0.9997	y = 1.719 × 10^7^x + 1.688 × 10^4^	0.9999	31.63	y = 1.197 × 10^7^x + 2.255 × 10^4^	0.9998	−72.92
Dimethoate	y = 2.781 × 10^6^x − 3.610 × 10^2^	0.9998	y = 1.709 × 10^6^x + 2.219 × 10^3^	0.9999	−38.55	y = 1.183 × 10^6^x + 4.957 × 10^3^	0.9998	−57.49
Diniconazole	y = 7.137 × 10^6^x + 7.200 × 10^2^	0.9999	y = 8.783 × 10^6^x + 5.000 × 10^4^	0.9999	23.06	y = 6.410 × 10^6^x + 9.208 × 10^3^	0.9997	−10.19
Epoxiconazole	y = 2.346 × 10^6^x − 1.861 × 10^3^	0.9997	y = 2.330 × 10^5^x + 5.003 × 10^3^	0.9993	−90.07	y = 1.942 × 10^6^x + 7.745 × 10^3^	0.9989	−17.24
Fenaminstrobin	y = 1.565 × 10^7^x − 2.214 × 10^3^	0.9999	y = 1.328 × 10^7^x + 1.433 × 10^4^	0.9999	−15.17	y = 9.219 × 10^6^x + 2.201 × 10^3^	0.9996	−41.11
Flumetsulam	y = 2.625 × 10^6^x + 8.190 × 10^2^	0.9999	y = 2.330 × 10^6^x + 1.969 × 10^3^	0.9997	11.27	y = 1.466 × 10^6^x + 4.800 × 10^3^	0.9997	−44.14
Hexaconazole	y = 5.566 × 10^6^x + 1.881 × 10^4^	0.9970	y = 7.227 × 10^6^x + 3.091 × 10^4^	0.9995	29.84	y = 5.627 × 10^6^x + 4.042 × 10^4^	0.9996	1.10
Imidacloprid	y = 5.180 × 10^5^x + 1.700 × 10^2^	0.9999	y = 4.162 × 10^5^x + 4.260 × 10^2^	0.9998	−19.65	y = 2.992 × 10^5^x + 3.506 × 10^3^	0.9989	−42.23
Isoproturon	y = 8.141 × 10^6^x − 1.327 × 10^3^	0.9998	y = 8.526 × 10^6^x + 7.472 × 10^3^	0.9998	4.72	y = 7.659 × 10^6^x − 6.180 × 10^2^	0.9997	−5.93
Kresoxim-methyl	y = 2.844 × 10^5^x − 1.000 × 10^2^	0.9999	y = 2.249 × 10^5^x + 3.520 × 10^2^	0.9996	−20.92	y = 9.482 × 10^4^x + 5.030 × 10^2^	0.9988	−66.66
Metsulfuron-methyl	y = 2.631 × 10^6^x + 2.208 × 10^3^	0.9996	y = 4.487 × 10^6^x + 5.776 × 10^3^	0.9999	70.51	y = 3.479 × 10^6^x + 5.900 × 10^1^	0.9997	32.20
Myclobutanil	y = 4.562 × 10^6^x + 1.899 × 10^3^	0.9999	y = 4.454 × 10^6^x + 2.797 × 10^3^	0.9999	−2.37	y = 3.891 × 10^6^x + 2.403 × 10^3^	0.9995	−14.71
Pirimicarb	y = 2.057 × 10^8^x + 1.849 × 10^5^	0.9998	y = 2.118 × 10^8^x + 9.480 × 10^4^	0.9998	2.97	y = 1.891 × 10^8^x + 5.004 × 10^4^	0.9998	−8.10
Tebuconazole	y = 8.610 × 10^6^x − 5.680 × 10^2^	0.9999	y = 8.456 × 10^6^x + 4.180 × 10^5^	0.9993	−1.79	y = 6.792 × 10^6^x + 1.716 × 10^5^	0.9996	−21.12
Thiamethoxam	y = 1.397 × 10^7^x + 6.632 × 10^4^	0.9960	y = 9.586 × 10^6^x + 1.303 × 10^4^	0.9993	−31.39	y = 5.613 × 10^6^x + 1.105 × 10^4^	0.9998	−59.82
Triazophos	y = 1.210 × 10^8^x + 3.927 × 10^5^	0.9968	y = 1.737 × 10^8^x + 2.675 × 10^5^	0.9991	43.47	y = 1.445 × 10^8^x + 1.431 × 10^5^	0.9994	19.42
Trichlorfon	y = 2.314 × 10^7^x − 9.036 × 10^4^	0.9949	y = 1.866 × 10^7^x + 2.171 × 10^4^	0.9998	−19.36	y = 1.281 × 10^7^x + 9.128 × 10^4^	0.9999	−44.63

**Table 3 foods-14-04351-t003:** Accuracy and precision of 26 pesticides in wheat grain and bran at three spiked levels (n = 5).

Pesticides	Wheat Grain						Wheat Bran					
	0.005 (mg kg^−1^)		0.1 (mg kg^−1^)		0.2 (mg kg^−1^)		0.01 (mg kg^−1^)		0.1 (mg kg^−1^)		0.2 (mg kg^−1^)	
	Recovery (%)	RSD (%)	Recovery (%)	RSD (%)	Recovery (%)	RSD (%)	Recovery (%)	RSD (%)	Recovery (%)	RSD (%)	Recovery (%)	RSD (%)
Acetamiprid	101.8	5.6	91.0	15.7	76.5	3.6	101.9	2.5	99.1	4.6	94.7	2.2
Azoxystrobin	104.4	5.0	98.1	9.7	86.3	8.8	107.7	2.7	103.5	4.6	104.4	0.9
Bensulfuron methyl	111.3	9.0	100.5	12.4	87.1	3.5	110.7	2.4	104.5	3.3	97.7	1.7
Chlortoluron	116.8	6.3	93.7	13.7	73.9	2.4	101.5	5.0	99.4	2.6	95.8	2.6
Clothianidin	97.5	10.7	95.2	14.6	80.1	3.5	93.9	9.7	102.6	3.3	95.2	2.8
Cyproconazole	102.6	9.6	81.2	5.9	85.3	1.8	104.2	1.9	101.4	2.6	93.9	2.1
Diazinon	98.9	4.3	88.4	3.8	81.9	2.8	94.1	4.7	95.3	2.4	87.7	4.1
Dichlorvos	103.6	8.3	81.9	6.7	92.4	6.8	97.6	7.1	94.7	18.5	95.8	2.4
Difenoconazole	102.1	9.1	96.6	11.7	81.8	2.1	101.1	3.4	101.4	3.7	96.9	2.7
Diflufenican	92.9	3.2	94.6	3.6	78.3	5.1	102.0	10.2	92.3	1.1	101.7	8.9
Dimethoate	99.6	3.5	86.6	12.9	77.2	1.1	98.4	3.7	100.8	3.6	92.8	2.2
Diniconazole	105.6	3.9	92.1	12.5	80.5	2.5	97.9	1.7	97.9	1.4	93.1	2.4
Epoxiconazole	108.8	4.3	109.3	2.4	84.7	2.1	107.2	0.8	103.6	1.6	98.4	2.7
Fenaminstrobin	107.4	6.9	88.5	6.5	84.1	3.1	102.7	4.1	97.8	4.6	93.1	5.8
Flumetsulam	101.7	8.9	92.3	11.8	73.2	4.8	98.4	3.9	99.3	2.4	91.7	2.1
Hexaconazole	101.9	4.4	103.2	9.9	83.1	3.7	93.0	2.9	96.3	3.0	93.6	1.9
Imidacloprid	98.3	6.2	110.7	17.1	85.8	4.3	105.9	8.1	105.4	6.7	98.5	3.6
Isoproturon	102.0	10.2	83.4	10.3	81.6	1.5	92.3	1.1	98.7	1.7	92.1	1.6
Kresoxim-methyl	101.6	9.5	108.6	17.9	88.4	5.5	99.0	10.7	99.4	3.0	94.5	4.2
Metsulfuron-methyl	102.8	6.6	88.5	6.6	88.2	4.9	105.1	4.2	101.7	1.8	97.3	3.3
Myclobutanil	105.5	7.2	98.6	7.7	89.4	2.3	104.5	3.2	97.9	4.5	101.2	3.8
Pirimicarb	106.3	6.9	86.4	2.9	102.1	2.2	103.2	3.9	103.7	1.0	95.5	1.9
Tebuconazole	97.6	8.6	100.9	3.0	92.8	4.2	103.1	4.3	99.6	4.1	93.8	3.3
Thiamethoxam	93.9	3.3	83.8	3.0	84.9	3.2	97.6	5.9	101.2	1.4	93.5	1.9
Triazophos	103.8	4.9	84.7	4.8	88.4	3.7	101.7	2.2	102.1	2.0	97.1	2.9
Trichlorfon	104.4	8.3	86.4	4.1	102.0	2.6	102.3	4.0	103.7	2.6	95.6	5.8

## Data Availability

The original contributions presented in the study are included in the article/Appendix A; further inquiries can be directed to the corresponding authors.

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
