# Peer review of "Simultaneous Determination and Dietary Risk Assessment of 26 Pesticide Residues in Wheat Grain and Bran Using QuEChERS-UHPLC-MS/MS"

_foods, 2025, doi:10.3390/foods14244351_

Round 1

Reviewer 1 Report

Comments and Suggestions for Authors

Dear authors,

Overall, the study is well structured and the findings are clearly presented. However, several revisions are necessary to further strengthen the scientific rigor of the manuscript. Detailed recommendations, including clarifications, reference additions, methodological adjustments, and formatting corrections, are provided in the attached PDF file. I recommend revising the manuscript in accordance with these suggestions.

Author Response

Response to Reviewer 1 Comments

Comments 1: Including long URL links directly in the main text is not consistent with standard journal formatting requirements. The reference to the FAO data should be provided in the reference list, including the web address, formatted according to the journal’s citation style. Please remove the URL from the text and revise the citation accordingly.

Response 1: Thank you for your valuable comments. We fully agree with your suggestion. Accordingly, we have removed all direct URLs from the manuscript and replaced them with citations in the journal's standard reference format. These revisions were located on Page 1, Lines 33 and 39.

Comments 2: This is a strong statement and requires support from appropriate references. Please provide reliable references to support this statement.

Response 2: Thanks for your suggestion. We have now incorporated citations to literature addressing the quality and safety implications of persistent residues in wheat. Please refer to Page 1, Line 41.

Comments 3: The manuscript would benefit from integrating additional recent and relevant literature. For example, the study ‘Pesticide Residues in Wheat Grains in Turkey (2021–2024): Multi-Year Monitoring and Health Risk Assessment’

(https://doi.org/10.1016/j.jfca.2025.108053) provides valuable multi-year monitoring data and a comprehensive risk assessment approach. I recommend that the authors consider including and discussing this reference to strengthen the context and comparative value of their work.

Response 3: Thanks for your help with our manuscript. We have already cited the papers you mentioned. Please refer to Page 2, Line 60-63. Furthermore, we have added two additional relevant publications to strengthen this section (Page 2, Lines 63–68). Yigitsoy et al. investigated pesticide residues in major wheat-producing regions of Turkey, finding that both chronic and acute health risk quotients (HQc and HQa) for adults and children were below the threshold of 1 [1]. In a study of 80 wheat samples from Algeria, Mebdoua et al. reported that 5% of samples exceeded the maximum residue limit (MRL) for benalaxyl, chlorpyrifos, and metalaxyl; despite this, the estimated long-term exposure for these pesticides remained below 100% of the acceptable daily intake (ADI) [2]. Similarly, Kalantary et al. examined wheat grains and their processed products from 12 fields in the Arak-Bidgol region of Iran, demonstrating that the total health risk indices for both children and adults were also below 1 [3].

[1] Yigitsoy, C.; Sadighfard, S. Pesticide residues in wheat grains in Türkiye (2021–2024): Multi-year monitoring and health risk assessment. Journal of Food Composition and Analysis 2025, 147, 108053, doi:https://doi.org/10.1016/j.jfca.2025.108053.

[2] Mebdoua, S.; Ounane, G. Evaluation of pesticide residues in wheat grains and its products from Algeria. Food Additives & Contaminants: Part B 2019, 12, 289-295, doi:https://doi.org/10.1080/19393210.2019.1661529.

[3] Kalantary, R.R.; Jaafarzadeh, N.; Kermani, M.; Hesami Arani, M. Deltamethrin and malathion pesticide residues determination in the wheat and probabilistic health risk assessment by Monte Carlo simulation: a case study in Aran-Bidgol, Iran. International Journal of Environmental Analytical Chemistry 2024, 104, 5701-5712, doi:https://doi.org/10.1080/03067319.2022.2128795.

Comments 4: While 2.50 g of wheat bran was used, the extraction solvent volume (5 mL water + 10 mL acetonitrile with 2% acetic acid) was kept the same as for wheat grain. This results in a different sample-to-solvent ratio for bran compared to grain.This results in a different sample-to-solvent ratio for bran compared to grain. The authors should clarify the rationale for this choice.

Response 4: Thank you for this insightful comment. The use of a reduced sample mass for wheat bran (2.5 g) compared to wheat grains was a deliberate methodological adjustment to ensure analytical accuracy and reproducibility. During our preliminary optimization, we determined that the target analytes were more concentrated in the bran matrix. Furthermore, wheat bran possesses a lower moisture content and a more porous structure than whole grains. Employing an identical sample mass for both matrices would result in analyte concentrations exceeding the instrument's linear range, thereby compromising quantitative accuracy and potentially leading to incomplete extraction. This adjustment is consistent with established practices in the field; for instance, similar reductions in sample mass for husks, bran, and straw are employed in methods for pesticide residue analysis in rice [4].

[4] Ashok Kumar Karedla, R. Surya Raj, S.V. Krishnamoorthy, A. Suganthi, K. Bhuvaneswari, S. Karthikeyan, P. Geetha, M. Senthilkumar, S. Jeyarajan Nelson,Validation, dissipation kinetics and monitoring of flonicamid and dinotefuran residues in paddy grain, straw, its processed produces and bran oil using LC-MS/MS. Food Chemistry, 2024, 435, 137589, doi:https://doi.org/10.1016/j.foodchem.2023.137589.

Comments 5: The chemical formula ‘C10H11ClN4’ should be formatted with numerical subscripts (e.g., C₁₀H₁₁ClN₄) according to standard chemical notation. Please revise the formula accordingly throughout the manuscript.

Response 5: Thank you for pointing this out. All chemical formulas within the manuscript have been meticulously reviewed and corrected. The updated information was presented in Table 1.

Comments 6: ...... are provided without any supporting reference. Please cite an appropriate source to justify these thresholds. For instance, https://doi.org/10.1016/j.foodcont.2022.109576.

Response 6: Thank you for your valuable comments. The cited references regarding matrix inhibition and enhancement effects have now been included in the manuscript to substantiate our arguments (Page 5, Line 160).

Comments 7: Formulas (2) and (3) used for calculating NEDI and HQc are presented without citing a source. These equations are based on standardized methodologies established by international authorities (e.g., JMPR/FAO/WHO, EFSA). Please provide the appropriate regulatory or scientific reference to support these calculations. The absence of citation introduces methodological uncertainty

Response 7: Thank you for pointing this out. We have supplemented the data and sources section of the manuscript with references and guidelines. Please refer to Page 6, Line 185-186, Page 6, Line 190.

Comments 8: The rationale for using the median residue (MRᵢ) is not explained. Median values are typically used for STMR data derived from supervised field trials, whereas the present study uses market/commercial samples. In such cases, mean residue values are more commonly applied.

Response 8: Thank you for pointing this out. In dietary risk assessment, the mean value is commonly used; in our experiment, some pesticides were not detected, but there are extreme cases. For example, tebuconazole and azoxystrobin were detected in only one or two samples at concentrations of 0.2924 mg kg-1 and 0.3925 mg kg-1, respectively. In other samples, their concentrations were below 0.2 mg kg-1. If we use the mean residue calculation, it may be affected by the extreme value. Risk assessment is to ensure public health and safety, and median residue values are typically below the highest residue but above the mean, allowing for a conservative estimation of exposure risk while, at the same time, ensuring that they avoid underestimating the risk because the mean is pulled down by a low residue sample. In addition, Wu et al,[5]. for pesticide residue detection of 56 pesticides and 21 metabolites in tea and for risk assessment, used the median residue to calculate long-term dietary risk. Wu et al.[6], also adopted the median residue to analyze the long-term dietary exposure risk in the health risk assessment of 8 vegetable species in Southwest China. If you insist on revising the median residue indicator to the mean residue, we will make the corresponding adjustments in the next revision.

  • Wu XQ, Li JX, Wei J, Tong KX, Xie YJ, Chang QY, Yu XX, Li B, L, ML, Fan CL, Chen H. Multi-residue analytical method development and dietary exposure risk assessment of 345 pesticides in mango by LC-Q-TOF/MS.Food Control, 2024, 170, 111016, dio: https://doi.org/10.1016/j.foodcont.2024.111016.
  • Wu YL, An QS, Li D, Kang L, Zhou CR, Zhang JB, Pan CP. Multi-residue analytical method development and risk assessment of 56 pesticides and their metabolites in tea by chromatography tandem mass spectroscopy. Food Chemistry, 2022, 375, 131819, doi: https://doi.org/10.1016/j.foodchem.2021.131819.

Comments 9: “HQc <1 and HQc ≥1” Reference ?

Response 9: Thanks for your suggestion. We have cited relevant reference. Please refer to Page 6, Line 190.

Comments 10: Since HQc is calculated as a percentage, the acceptable risk criterion should be expressed as ‘%HQc < 100’. To avoid confusion with the conventional ‘HQc < 1’ approach, the threshold should be revised and clearly stated in the Methods section

Response 10: Thanks for your careful checks. We apologize for the oversight. Based on your comments, we have corrected the text to ensure unit consistency throughout the manuscript. The change can be found on Page 6, Line 190.

Reviewer 2 Report

Comments and Suggestions for Authors

The manuscript presents a technically sound analytical method and produces valuable data on pesticide residues in wheat grain and bran in China. However, the manuscript needs revision before it is suitable for publication.

- The manuscripts content many editorial issues should be corrected

At section 3.4 in the caption of Figure 5 (page 13) is mis numbered as “Figure 2. Chronic dietary risk of pesticide bran…”. This should be corrected and made consistent with the order in the text.

At Section 2.5, the third subsection is labelled “5.5.3. Matrix effect” instead of “2.5.3”.

Ensure consistent figure numbering, currently Figure 3 is on page 11, Figure 4 on page 12, and the bran risk plots on page13 are again labelled “Figure 2”.

The tables and figures; Figure 1 and Figure 2 (page 7), Figure 3 (page 11) and Table 3 (page 9) should be revised.

- The abbreviations are defined at first use (e.g., HQc, NEDI, ADI, ME, MRL) must be check.

- For the selection of 26 pesticides in the introduction indicates many authorised formulations on wheat, but the rationale for choosing specifically these 26 active substances is not clearly articulated.

- In the section 2.6, states that samples were collected from 12 regions and briefly describes preparation, but does not specify (e.g: harvest year(s), whether samples were collected at farm level, storage facilities, or retail markets; number of samples per region; or criteria for region selection) please provide more informations about the origin and timing of samples.

- Only chronic exposure is assessed, although some of the insecticides (e.g., dimethoate, dichlorvos) may have relevant acute toxicity endpoints, please provide more discussion about these cases.

Author Response

Comments 1: At section 3.4 in the caption of Figure 5 (page 13) is mis numbered as “Figure 2. Chronic dietary risk of pesticide bran…”. This should be corrected and made consistent with the order in the text.

Response 1: Thank you for your valuable comments. The problem has been corrected. Please refer to Figure 5.

Comments 2: At Section 2.5, the third subsection is labelled “2.5.3. Matrix effect” instead of “5.5.3”.

Response 2: Thank you for pointing this out. “5.5.3” has been changed to “2.5.3”. Please refer to Page 5, Line 154.

Comments 3: Ensure consistent figure numbering, currently Figure 3 is on page 11, Figure 4 on page 12, and the bran risk plots on page13 are again labelled “Figure 2”.

Response 3: Thank you for your observation. The figure citation in the text has been corrected; please refer to Figure 5 on page 13.

Comments 4: The tables and figures; Figure 1 and Figure 2 (page 7), Figure 3 (page 11) and Table 3 (page 9) should be revised.

Response 4: Thanks for your suggestion. Figures 1, 2, and 3, along with Table 3, have been revised accordingly. The updated versions can be found on pages 7–8, 12, and 9–10, respectively.

Comments 5: The abbreviations are defined at first use (e.g., HQc, NEDI, ADI, ME, MRL) must be check.

Response 5: Thanks for your suggestion. The manuscript has been revised to include the full term upon first use for all abbreviated terms, as recommended. These changes can be found on Page 5, Lines 137, 148, and 154.

Comments 6: For the selection of 26 pesticides in the introduction indicates many authorised formulations on wheat, but the rationale for choosing specifically these 26 active substances is not clearly articulated.

Response 6: The 26 target pesticides selected for this study were all registered for use in wheat cultivation in China (China Pesticide Information Network). As a major global food crop, wheat is frequently threatened by pests, diseases, and weeds, which led to significant yield losses. Pesticide remains the most cost-effective and widely adopted method for managing these threats in modern agriculture. A review of the literature confirms that these 26 pesticides were commonly applied to control major wheat pests (e.g., wheat aphids), diseases (e.g., Fusarium head blight and wheat stripe rust), and weeds (e.g., Alopecurus aequalis and Descurainia sophia) [1, 2]. Furthermore, numerous studies have reported frequent detections of these pesticides in wheat grains, straw, bran, and soil, highlighting concerns regarding their environmental persistence and potential ecological risks [3–5]. Consequently, the selection of these pesticides is justified by their registered status, prevalence in agricultural practice, and the documented occurrence of their residues.

[1] MacDonald, GE; Kanissery, RG; Devkota, P; AF Daramola, Olumide S.; MacDonald, Gregory E.; Kanissery, Ramdas G.;Devkota, Pratap. TI Effects of co-applied agrochemicals on herbicide performance: A review. Crop Protection 2023, 174, 106396, doi:https://doi.org/10.1016/j.cropro.2023.106396.

[2] Cuicui Wang, Wenyang Dong, Jiao Shang, Hongbao Li, Zhao Chen, Bin Zhu, Pei Liang, Xueyan Shi, S431F mutation on AChE1 and overexpression of P450 genes confer high pirimicarb resistance in Sitobion miscanthi, Pesticide Biochemistry and Physiology, 2024, 202, 105957, doi:https://doi.org/10.1016/j.pestbp.2024.105957. 

[3] Rafique, N., Tariq, S.R. & Ahmed, D. Monitoring and distribution patterns of pesticide residues in soil from cotton/wheat fields of Pakistan. Environ Monit Assess, 2016, 188, doi:https://doi.org/10.1007/s10661-016-5668-6.

[4] Elena Hakme, Parvaneh Hajeb, Susan Strange Herrmann, Mette Erecius Poulsen, Processing factors of pesticide residues in cereal grain fractions, Food Control, 2024, 161, 110369, doi:https://doi.org/10.1016/j.foodcont.2024.110369.

[5] Słowik-Borowiec, M.; Szpyrka, E.; Książek-Trela, P.; Podbielska, M. Simultaneous Determination of Multi-Class Pesticide Residues and PAHs in Plant Material and Soil Samples Using the Optimized QuEChERS Method and Tandem Mass Spectrometry Analysis. Molecules, 2022, 27, 2140. https://doi.org/10.3390/molecules27072140.

Comments 7: In the section 2.6, states that samples were collected from 12 regions and briefly describes preparation, but does not specify (e.g: harvest year(s), whether samples were collected at farm level, storage facilities, or retail markets; number of samples per region; or criteria for region selection) please provide more informations about the origin and timing of samples.

Response 7: Thanks for your suggestion. Wheat grain samples were collected from 12 distinct regions across the country, alongside wheat bran samples from 11 regions; all collections were conducted in the 2024 growing season. The grain samples were obtained directly from farmers' fields, while the bran samples were sourced from local milling facilities. This specific methodological detail has now been incorporated into the manuscript. Please refer to Page 5, Line 163-164 and 168-178.

Comments 8: Only chronic exposure is assessed, although some of the insecticides (e.g., dimethoate, dichlorvos) may have relevant acute toxicity endpoints, please provide more discussion about these cases.

Response 8: Thank you for your suggestion. The manuscript has been revised to address these points; please refer to the added content on page 13, lines 386–390.

Organophosphorus pesticide (OPP) are highly toxic and bioaccumulative and pose a serious threat to human and wildlife health[6]. In larval zebrafish, moderate OPP exposure is associated with muscle paralysis and reduced heart rate, while severe exposure can also reduce trunk length [7]. Despite these risks, an analysis of dietary OPP residues in the Hungarian population found that over 99.95% of the estimated daily intake was below the acute reference dose [8].

[6] Kosimov, D.; Ergashev, R.; Mavjudova, A.; Kuziev, S. Organophosphorus Pesticide Degradation by Microorganisms: A Review. Frontiers in bioscience (Elite edition) 2025, 17, 38805, doi:https://doi.org/10.31083/fbe38805.

[7] Koenig, J.A.; Acon Chen, C.; Shih, T.-M. Development of a Larval Zebrafish Model for Acute Organophosphorus Nerve Agent and Pesticide Exposure and Therapeutic Evaluation. Toxics 2020, 8, 106, doi:https://doi.org/10.3390/toxics8040106.

[8] Zentai, A.; Szabó, I.J.; Kerekes, K.; Ambrus, Á. Risk assessment of the cumulative acute exposure of Hungarian population to organophosphorus pesticide residues with regard to consumers of plant based foods. Food and Chemical Toxicology 2016, 89, 67-72, doi:https://doi.org/10.1016/j.fct.2016.01.016.

Reviewer 3 Report

Comments and Suggestions for Authors

Introduction:

  1. Can you explain what was the reason for selection of those 26 pesticides? Are they authorised in China, are they authorised on cereals,…? Please insert clarification in paper.

Materials and methods

  1. Purity >95 % for standards is on the limit of acceptable. Certified standards usually have exact purity (for instance 99%).
  2. For sample collection you can insert table with data about no. of wheat grain and wheat bran samples per each of 12 regions

Results and discussion

  1. Please report how many calibration points were included in linear calibration curves?
  2. Considering that some MRLs were exceeded you should conduct acute risk assessment - whether these samples pose a short-term risk for consumers (with one meal). Please insert acute risk assessment results in paper.

Suplement

  1. Please insert in supplement median residue levels and ARfDs.

Author Response

Response to Reviewer 3 Comments

Comments 1: Can you explain what was the reason for selection of those 26 pesticides? Are they authorised in China, are they authorised on cereals,…? Please insert clarification in paper.

Response 1: The 26 target pesticides selected for this study were all registered for use in wheat cultivation in China (China Pesticide Information Network). As a major global food crop, wheat is frequently threatened by pests, diseases, and weeds, which led to significant yield losses. Pesticide remains the most cost-effective and widely adopted method for managing these threats in modern agriculture. A review of the literature confirms that these 26 pesticides were commonly applied to control major wheat pests (e.g., wheat aphids), diseases (e.g., Fusarium head blight and wheat stripe rust), and weeds (e.g., Alopecurus aequalis and Descurainia sophia) [1, 2]. Furthermore, numerous studies have reported frequent detections of these pesticides in wheat grains, straw, bran, and soil, highlighting concerns regarding their environmental persistence and potential ecological risks [3–5]. Consequently, the selection of these pesticides is justified by their registered status, prevalence in agricultural practice, and the documented occurrence of their residues.

[1] MacDonald, GE; Kanissery, RG; Devkota, P; AF Daramola, Olumide S.; MacDonald, Gregory E.; Kanissery, Ramdas G.;Devkota, Pratap. TI Effects of co-applied agrochemicals on herbicide performance: A review. Crop Protection 2023, 174, 106396, doi:https://doi.org/10.1016/j.cropro.2023.106396.

[2] Cuicui Wang, Wenyang Dong, Jiao Shang, Hongbao Li, Zhao Chen, Bin Zhu, Pei Liang, Xueyan Shi,

S431F mutation on AChE1 and overexpression of P450 genes confer high pirimicarb resistance in Sitobion miscanthi, Pesticide Biochemistry and Physiology, 2024, 202, 105957, doi:https://doi.org/10.1016/j.pestbp.2024.105957.

[3] Rafique, N., Tariq, S.R. & Ahmed, D. Monitoring and distribution patterns of pesticide residues in soil from cotton/wheat fields of Pakistan. Environ Monit Assess, 2016, 188, doi:https://doi.org/10.1007/s10661-016-5668-6.

[4] Elena Hakme, Parvaneh Hajeb, Susan Strange Herrmann, Mette Erecius Poulsen, Processing factors of pesticide residues in cereal grain fractions, Food Control, 2024, 161, 110369, doi:https://doi.org/10.1016/j.foodcont.2024.110369.

[5] Słowik-Borowiec, M.; Szpyrka, E.; Książek-Trela, P.; Podbielska, M. Simultaneous Determination of Multi-Class Pesticide Residues and PAHs in Plant Material and Soil Samples Using the Optimized QuEChERS Method and Tandem Mass Spectrometry Analysis. Molecules, 2022, 27, 2140. https://doi.org/10.3390/molecules27072140.

Comments 2: Purity >95 % for standards is on the limit of acceptable. Certified standards usually have exact purity (for instance 99%).

Response 2: Thank you for your valuable comments. The standards used in this study were mixed standards. The purity of each individual standard is detailed in the revised manuscript on Page 3, Line 90.

Pesticides

purity (%)

Pesticides

purity (%)

Pesticides

purity (%)

Pesticides

purity (%)

Acetamiprid

99.8

Cyproconazole

99.9

Dimethoate

99.9

Hexaconazole

99.9

Azoxystrobin

99.6

Diazinon

99.8

Diniconazole

99.0

Imidacloprid

99.8

Bensulfuron methyl

98.2

Dichlorvos

99.3

Epoxiconazole

99.9

Isoproturon

99.1

Chlorotouron

99.8

Difenoconazole

99.4

Fenaminstrobin

99.9

Kresoxim-methyl

99.9

Clothianidin

99.8

Diflufenican

99.9

Flumetsulam

99.9

Metsulfuron-methyl

98.3

Myclobutanil

99.8

Pirimicarb

99.5

Tebuconazole

99.8

Thiamethoxam

99.6

Triazophos

99.8

Trichlorfon

99.0

Comments 3: For sample collection you can insert table with data about no. of wheat grain and wheat bran samples per each of 12 regions.

Response 3: Thanks for your suggestion. Wheat grain samples were collected from 12 distinct regions across the country, while wheat bran samples were obtained from 11 regions; all collections were conducted in the 2024 growing season. The grain samples were acquired directly from farmers' fields, and the bran samples were sourced from local milling facilities. This specific information regarding the sample collection methodology has now been incorporated into the manuscript. Please refer to Page 5, 163-164 and 168-178.

Comments 4: Please report how many calibration points were included in linear calibration curves?

Response 4: Thank you for pointing this out. A total of nine calibration points were used for the calibration curves, with the following concentrations: wheat grain (0.001, 0.002, 0.0025, 0.005, 0.01, 0.02, 0.05, 0.1, 0.2 mg kg-1) and wheat bran (0.002, 0.0025, 0.005, 0.01, 0.02, 0.025, 0.05, 0.1, 0.2 mg kg-1). Please refer to Page 5, Line 143-146.

Comments 5: Considering that some MRLs were exceeded you should conduct acute risk assessment - whether these samples pose a short-term risk for consumers (with one meal). Please insert acute risk assessment results in paper.

Response 5: Thank you for this suggestion. In response, we have incorporated a section on acute risk assessment. An acute dietary risk assessment was performed for all pesticides detected in wheat grains and bran that have an established acute reference dose (ARfD). The results demonstrate that the acute hazard quotient (HQa) values for all target pesticides were substantially below 100%. This indicates that the short-term dietary risk is acceptable and is unlikely to pose adverse health effects to populations across different age groups. For details, please see Page 6, Lines 191–198, and Page 13, Lines 372–395.

Comments 6: Please insert in supplement median residue levels and ARfDs.

Response 6: Thanks for your suggestion. we have supplemented the relevant content. Please refer to supplement Table S1.

Round 2

Reviewer 3 Report

Comments and Suggestions for Authors

Dear Authors!

I am finding your revision acceptable and I am recommending publication.